# Deep Learning Methodologies Applied to Digital Pathology in Prostate Cancer: A Systematic Review

**DOI:** 10.3390/diagnostics13162676

**Published:** 2023-08-14

**Authors:** Noémie Rabilloud, Pierre Allaume, Oscar Acosta, Renaud De Crevoisier, Raphael Bourgade, Delphine Loussouarn, Nathalie Rioux-Leclercq, Zine-eddine Khene, Romain Mathieu, Karim Bensalah, Thierry Pecot, Solene-Florence Kammerer-Jacquet

**Affiliations:** 1Impact TEAM, Laboratoire Traitement du Signal et de l’Image (LTSI) INSERM, Rennes University, 35033 Rennes, Francesoleneflorence.kammerer-jacquet@chu-rennes.fr (S.-F.K.-J.); 2Department of Pathology, Rennes University Hospital, 2 rue Henri Le Guilloux, CEDEX 09, 35033 Rennes, France; pierre.allaume@chu-rennes.fr (P.A.);; 3Department of Radiotherapy, Centre Eugène Marquis, 35033 Rennes, France; 4Department of Pathology, Nantes University Hospital, 44000 Nantes, France; 5Department of Urology, Rennes University Hospital, 2 rue Henri Le Guilloux, CEDEX 09, 35033 Rennes, France; 6Facility for Artificial Intelligence and Image Analysis (FAIIA), Biosit UAR 3480 CNRS-US18 INSERM, Rennes University, 2 Avenue du Professeur Léon Bernard, 35042 Rennes, France

**Keywords:** prostate cancer, deep learning, digital pathology, Gleason grading, convolutional neural networks, artificial intelligence

## Abstract

Deep learning (DL), often called artificial intelligence (AI), has been increasingly used in Pathology thanks to the use of scanners to digitize slides which allow us to visualize them on monitors and process them with AI algorithms. Many articles have focused on DL applied to prostate cancer (PCa). This systematic review explains the DL applications and their performances for PCa in digital pathology. Article research was performed using PubMed and Embase to collect relevant articles. A Risk of Bias (RoB) was assessed with an adaptation of the QUADAS-2 tool. Out of the 77 included studies, eight focused on pre-processing tasks such as quality assessment or staining normalization. Most articles (*n* = 53) focused on diagnosis tasks like cancer detection or Gleason grading. Fifteen articles focused on prediction tasks, such as recurrence prediction or genomic correlations. Best performances were reached for cancer detection with an Area Under the Curve (AUC) up to 0.99 with algorithms already available for routine diagnosis. A few biases outlined by the RoB analysis are often found in these articles, such as the lack of external validation. This review was registered on PROSPERO under CRD42023418661.

## 1. Introduction

### 1.1. Prostate Cancer

Prostate cancer (PCa) is one of the most prevalent cancers among male cancers, especially aging [1]. The gold standard for diagnosing and treating patients is the analysis of H&E histopathology slides [2]. The observation of the biopsy tissue enables pathologists to detect tumor cells and characterize the aggressiveness of the tumor using Gleason grading. This grading is based on gland structure and ranks 1 to 5 according to the differentiation [3]. When more than one pattern is present on the biopsy, the scoring is defined by the most represented pattern (primary) and the highest one (secondary). For instance, a biopsy with most of pattern 3 and some patterns 4 and 5 will be scored 3 + 5 = 8. To improve the prognosis correlation among these scores, an update of Gleason grading called the ISUP (International Society of Urological Pathology) Grading system, was proposed in 2014, which assigns patients into a group depending on the Gleason score (see Table 1) [4]. These groups have varying prognoses, from group 1 corresponding to an indolent tumor to group 5 having a poor prognosis [5].

### 1.2. DL Applied to WSI

Glass slides are digitized with scanners to form whole slide images (WSIs). These images can be used to train DL algorithms. The principle is to teach an algorithm to correlate WSIs to a target provided by the user called the ground truth (GT). Once training is over, it is essential to test (or validate) the algorithm on other WSIs to validate it on other images. All studies must therefore have a training and a testing (or validation) cohort. Additionally, to comprehensively evaluate a model, it is common to use cross-validation (cross-val), i.e., splitting the training dataset into *n* parts, called folds, to train a model with (*n* − 1)/*n* of all data and evaluate it with 1/*n* of all data, *n* times. Finally, the algorithm can be evaluated on an external dataset.

Two main types of algorithms are discussed in this paper: segmentation and classification algorithms. Segmentation algorithms aim at precisely delineating regions of interest in WSIs (see Figure 1A). For instance, a common segmentation task is to localize stroma and epithelium. The most popular architecture is U-Net [6]. Classification algorithms intend to estimate labels, called classes, associated with images. In pathomics, WSIs are divided into tiles that are then encoded into features representing a summary of the tiles. A classification process is then learned from the features (see Figure 1B). These features are deep-learned with convolutional neural networks (CNN). The main architectures that are used for classification are Inception [7], ResNet [8], VGG [9], DenseNet [10], and MobileNet [11]. Those classic architectures can be trained and customized, but it is also possible to use models pre-trained on other data and fine-tune these models on the desired data. This is called transfer learning.

A significant difficulty in pathomics is the very large size of WSIs which prevents them from being processed all at once. This requires tiling WSIs in many tiles and dealing with data of different sizes. Most architectures need the same size of input for every image. The number of tiles selected must therefore be the same for each image. The magnification at which the image is tiled is also a key parameter in handling the input data. Additionally, this leads to potentially very large amounts of data to manually annotate to create the GT, a time-consuming task for pathologists. Strategies such as weak supervision, where only one label per patient is assigned from the pathologists’ report, have emerged to speed up this work.

### 1.3. Applications and Evaluation of Algorithms

In pathomics applied to PCa, DL algorithms are applied for pre-processing, diagnosis, or prediction. Pre-processing algorithms evaluate input data quality and staining normalization. Diagnosis methods focus either on cancer detection or on Gleason grading. Prediction approaches focus on predicting survival and cancer progression or genomic signatures. Algorithms are evaluated with different metrics that are summarized in Appendix A. The most used metric is the Area Under the Curve (AUC) which can be used to evaluate diagnosis and prognosis tasks.

### 1.4. Aim of This Review

This paper reviews the main articles mentioning DL algorithms applied to prostate cancer WSIs until 2022. It highlights current trends, future perspectives, and potential improvements in the field. It aims to be a thorough but comprehensive take on this subject for everyone interested.

## 2. Materials and Methods

This systematic review followed the PRISMA guidelines, containing advices and checklists to frame systematic reviews [12]. It was registered on PROSPERO, a website identifying all existing and undergoing systematic reviews under CRD42023418661.

PubMed and Embase, two biomedical databases, were used to look for articles until 1 September 2023 with the following keywords:

(‘prostate’ OR prostate OR prostatic) AND (cancer OR carcinoma OR malignant OR lesions) AND (‘artificial intelligence’ OR ‘algorithm’ OR ‘deep learning’ OR ‘machine learning’ OR ‘automated classification’ OR ‘supervised learning’ OR ‘neural network’) AND (‘whole slide image’ OR ‘digital pathology’ OR pathomics OR ‘he’ OR ‘H&E’ OR ‘histological’)

Additionally, selected papers had to:be written in English,focus on prostate cancer,use pathology H&E-stained images,rely on deep learning.

The selection of papers was first evaluated with titles only. Abstracts were then reviewed, leading to a collection of 45 selected papers. Finally, 32 articles that did not come through the database search but were referenced in many of the 45 papers were manually added. Additional searches via PubMed and Google were performed to check if conference abstracts led to published articles. Figure 2 illustrates the selection process.

A risk-of-bias study was performed for each paper, using an adaptation of QUADAS-2 [13] (see Table 2) as the most suitable tool for AI papers. QUADAS-AI is not yet available [14]. Remarks made in this paper and criteria in the IJMEDI checklist [15] were used to create a homemade checklist adapted to this review. This allowed us to evaluate all papers on the same criteria to outline the main biases.

## 3. Results

Data was collected for every article: first author name, year of publication, aim of the paper, neural network used, number of slides in the training, validation (internal and external) cohorts, sub-aims of the paper and their corresponding performances with an Excel Sheet. All tables can be found in the Appendix A with more detailed information (Appendix A).

### 3.1. Pre-Processing

Out of the 77 selected articles, three proposed methods for WSI quality assessment and 6 attended to correct for staining and/or scanning discrepancies across a set of WSIs (Table 3).

#### 3.1.1. Quality Assessment

Two articles proposed to evaluate the quality of a collection of slides by using a purity score (AUC of 0.8) [18] or a binary criterion of usability (AUC of 0.983) [17], attributed by pathologists. A third study proposed to artificially alter a collection of WSIs with artifacts and to evaluate the resulting impact on the tissue classification of tiles, which was perfectly performed without artifacts (F1-score = 1) [16]. The authors demonstrated that the main causes for the drop in performance are defocusing, jpg compression and staining variations. Indeed, the latter topic has been the subject of quite some articles in the field.

#### 3.1.2. Staining Normalization and Tile Selection

WSIs stained in different locations show differences in intensity observed in the three-color channels (RGB). The scanners also influence the acquired WSIs. Many methods were proposed to normalize staining and/or scanning to obtain more generic DL models. Swiderska-Chadaj et al. evaluated the performance of their WSI normalization when classifying patients for benign/malignant tissues [24]. First, they improved classification performance with external datasets when images were scanned with the same scanner as the one used to scan the training cohort. Then, they trained a generative model to normalize WSIs for the external dataset, improving their classification performance (from an AUC of 0.92 to 0.98). Similarly, Rana et al. virtually stained with H&E non-stained WSIs with the help of generative models. Then, they stained the same biopsies with H&E and compared virtual and stained slides, reaching a high correlation (PC of 0.96) [22]. Salvi et al. focused not only on staining normalization but also on tile selection. Basing itself on segmentation, it allows for better select tiles to represent the WSI and its tissue’s complexity [25]. The model improvement (gain of 0.3 in specificity, reaching a sensitivity of 1) highlights the need for preprocessing steps, such as stain normalization or segmentation-guided tile selection.

### 3.2. Diagnosis

Most of the selected papers (*n* = 53) focused on diagnosis, whether it be about tissue segmentation (*n* = 5), detection of cancer tissue (*n* = 21), attribution of a Gleason grading (*n* = 10), or both detection of cancer tissue and Gleason grading (*n* = 19 with 2 in segmentation also). Diagnosis can be performed at the pixel level, tile level, WSI level or patient level. Methods processing data at the pixel level are segmentation algorithms trained to localize areas of clinical interest, e.g., malignant and benign epithelium. Methods processing data at tile, WSI or patient level are classification algorithms trained to identify categories of clinical interest associated with the input data.

#### 3.2.1. Segmentation

Many segmentation studies (see Table 4) demonstrated high performance for gland segmentation: a Dice of 0.84 [26], a Dice of 0.9 [27], an F1 score from 0.83 to 0.85 [28] and an AUC of 0.99 [29]. Precisely annotating WSIs is time-consuming, and several approaches focused on strategies to reduce the number of annotations. In [30], the authors trained a model with rough annotations obtained with traditional image processing techniques. Then they fine-tuned this model with a few precise annotations made by pathologists (AUC gain of 0.04). In [28], slides were first stained with immunohistochemistry (IHC) for which DL models were already trained. The slides were stained with H&E, and a new model was trained using the annotations masks obtained from the IHC segmentation model.

#### 3.2.2. Cancer Detection

Even if segmentation helps select tiles of interest, other processes can be used to improve classification. There is the possibility of using multi-scale embedding to take in an area at different magnifications at the same time (as a pathologist would do) [31]. To select tiles and improve the explainability of the model, one of the most popular approaches for patient classification in pathology is Multiple Instance Learning (MIL) [32,33,34]). From a WSI divided into a high number of tiles, the goal is to associate to each tile a probability for the considered classification task, e.g., the presence of cancer, and then use the most predictive tiles to diagnose. This DL approach directly addresses the tile selection during the training, allowing us to deal with WSIs of different sizes and identify the tiles used to decide. This paradigm was notably used for cancer detection by Campanella et al. on the largest training and external validation cohorts to date [32] and by Pinckaers et al. [33]. These articles achieve an AUC of 0.99 on internal validation cohorts and above 0.90 on external cohorts. Indeed, many other approaches obtained AUC over 0.95 [31,32,33,34,35,36,37,38,39,40,41,42,43,44,45,46]. In 2021, the FDA approved the deployment of PaigeProstate [34], based on the MIL architecture of Campanella [32]. It was evaluated with three different external cohorts [47,48,49]. It showed a high performance (sensitivity of minimum 0.94 and specificity of minimum 0.93), focusing on a high sensitivity to avoid missing small areas of cancer. Other articles focused on the detection of more specific patterns, such as cribriform patterns with an accuracy of 0.88 and AUC of 0.8 [50,51,52] or perineural invasion with an AUC of 0.95 [36]. To evaluate the generalization of their model trained on biopsies for which they obtained an AUC of 0.96, Tsuneki et al. applied it to TUR-P (TransUrethral Resection of Prostate) biopsies with an AUC of 0.80. After fine-tuning their model with colon WSIs, performance on TUR-P biopsies increased to 0.85 AUC [53]. Wen fine-tuning their model with the TUR-P training cohort, they obtained an AUC of 0.9 for every testing cohort [54]. All the articles focusing on cancer detection are in Table 5.

#### 3.2.3. Gleason Grading

When estimating Gleason grading, many papers only focused on classifying tiles or small regions like TMAs by taking advantage of classical CNN architectures trained on large datasets of natural images such as ImageNet [60]. In that context, tiles were encoded into features which corresponded to the input data for classification [21,38,61,62,63,64,65,66]. One of the first papers in the field used a cohort of 641 TMAs, obtaining a quadratic Cohen Kappa of 0.71 [21]. Similarly, Kott et al. obtained an AUC of 0.82 [62]. A few articles directly addressed the localization at the pixel level of Gleason patterns [28,29,41,67,68,69,70,71]. Performances vary between IoU of 0.48 [70,71] to IoU around 0.7 [28,29,68], Dice score of 0.74 [69], quadratic Cohen kappa of 0.854 [41], sensitivity of 0.77 and specificity of 0.94 [67] (see Table 6). Adding epithelium detection greatly improved performance when properly segmenting areas depending on Gleason grades (gain of 0.07 in mean IoU) [29]. Many articles used the same pipeline to group Gleason scores according to ISUP recommendations [35,36,52,72,73]. These studies relied on the annotation of glands according to their Gleason pattern. WSIs were then split into tiles, and every single tile was classified according to the majority Gleason pattern (GP). Once all tiles were classified, heatmaps were generated, and a classifier was trained to properly aggregate ISUP grade groups or at least differentiate high from low grade [74]. Two algorithms based on this pipeline are now commercially available: (i) IBEX [36] (AUC of 0.99 for cancer detection, AUC of 0.94 for low/high-grade classification); (ii) DeepDx [72] (quadratic kappa of 0.90). DeepDx was further evaluated on an external cohort by Jung et al. [75] (kappa of 0.65 and quadratic kappa of 0.90). A third algorithm capable of Gleason grading (kappa of 0.77) was commercialized by Aiforia [42]. Another important milestone in the field was the organization and the release of the PANDA challenge focusing on Gleason grading at the WSI level without gland annotations [76]. This is an incredibly large cohort (around 12,000 biopsies of 3500 cases) publicly available, including slides from 2 different locations and external validation from 2 other sites. Best algorithms reached a quadratic Kappa of 0.85 on external validation datasets. One of the goals of Gleason grading algorithms is the potential decrease of inter-observer variability for pathologists using the algorithm. Some algorithms already have better Kappas than a cohort of pathologists compared to the ground truth [21,41,77].

**Table 6 diagnostics-13-02676-t006:** Articles focusing on Gleason grading.

First Author, Year Reference	DL Architecture	Training Cohort	IV Cohort	EV Cohort	Aim	Results
Källén, 2016 [63]	OverFeat	TCGA	10-fold cross val	213 WSIs	GP tile classification	ACC: 0.81
					Classify WSIs with a majority GP	ACC: 0.89
Jimenez Del Toro, 2017 [74]	GoogleNet	141 WSIs	47 WSIs	None	High vs. Low-grade classification	ACC:0.735
Arvaniti, 2018 [21]	MobileNet + classifier	641 TMAs	245 TMAs	None	TMA grading	qKappa: 0.71/0.75 (0.71 pathologists)
					Tile grading	qKappa: 0.55/0.53 (0.67 pathologists)
Poojitha, 2019 [64]	CNNs	80 samples	20 samples	None	GP estimation at tile level (GP 2 to 5)	F1: 0.97
Nagpal, 2019 [73]	InceptionV3	1159 WSIs	331 WSIs	None	GG classification	ACC: 0.7
					High/low-grade classification (GG 2, 3 or 4 as threshold)	$\bar{AUC}$: 0.95
					Survival analysis, according to Gleason	HR: 1.38
Silva-Rodriguez, 2020 [52]	Custom CNN	182 WSIs	5-fold cross-val	703 tiles from 641 TMAs [21]	GP at the tile level	$\bar{F1}$: 0.713 (IV) & 0.57 (EV) qKappa: 0.732 (IV) & 0.64 (EV)
					GG at the WSI level	qKappa: 0.81 (0.77 with [21] method)
					Cribriform pattern detection at the tile level	AUC: 0.822
Otalora, 2021 [65]	MobileNet-based CNN	641 TMAs 255 WSIs	245 TMAs 46 WSIs	None	GG classification	wKappa: 0.52
Hammouda, 2021 [78]	CNNs	712 WSIs	96 WSIs	None	GP at the tile level	$\bar{F1}$: 0.76
					GG	$\bar{F1}$: 0.6
Marini, 2021 [66]	Custom CNN	641 TMAs 255 WSI	245 TMAs 46 WSIs	None	GP at the tile level	qKappa: 0.66
					GS at TMA level	qKappa: 0.81
Marron-Esquivel, 2023 [77]	DenseNet121	15,020 patches	2612 patches	None	Tile-level GP classification	qKappa: 0.826
		102,324 patches (PANDA + fine-tuning)				qKappa: 0.746
Ryu, 2019 [72]	DeepDx Prostate	1133 WSIs	700 WSIs	None	GG classification	qKappa: 0.907
Karimi, 2019 [61]	Custom CNN	247 TMAs	86 TMAs	None	Malignancy detection at the tile level	Sens: 0.86 Spec: 0.85
					GP 3 vs. 4/5 at tile level	Sens: 0.82 Spec: 0.82
Nagpal, 2020 [35]	Xception	524 WSIs	430 WSIs	322 WSIs	Malignancy detection at the WSI level	AUC: 0.981 Agreement: 0.943
					GG1-2 vs. GG3-5	AUC: 0.972 Agreement: 0.928
Pantanowitz, 2020 [36]	IBEX	549 WSIs	2501 WSIs	1627 WSIs	Cancer detection at the WSI level	AUC: 0.997 (IV) & 0.991 (EV)
					Low vs. high grade (GS 6 vs. GS 7–10)	AUC: 0.941 (EV)
					GP3/4 vs. GP5	AUC: 0.971 (EV)
					Perineural invasion detection	AUC: 0.957 (EV)
Ström, 2020 [37]	InceptionV3	6935 WSIs	1631 WSIs	330 WSIs	Malignancy detection at the WSI level	AUC: 0.997 (IV) 0.986 (EV)
					GG classification	Kappa: 0.62
Li, 2021 [39]	Weakly supervised VGG11bn	13,115 WSIs	7114 WSIs	79 WSIs	Malignancy of slides	AUC: 0.982 (IV) & 0.994 (EV)
					Low vs. high grade at the WSI level	Kappa: 0.818 Acc: 0.927
Kott, 2021 [62]	ResNet	85 WSIs	5-fold cross-val	None	Malignancy detection at the tile level	AUC: 0.83 ACC: 0.85 for fine-tuned detection
					GP classification at the tile level	Sens: 0.83 Spec: 0.94
Marginean, 2021 [79]	CNN	698 WSIs	37 WSIs	None	Cancer area detection	Sens: 1 Spec: 0.68
					GG classification	$\bar{Kappa}$: 0.6
Jung, 2022 [75]	DeepDx Prostate	Pre-trained	Pre-trained	593 WSIs	Correlation with reference pathologist (pathology report comparison)	Kappa: 0.654 (0.576) qKappa: 0.904 (0.858)
Silva-Rodriguez, 2022 [40]	VGG16	252 WSIs	98 WSIs	None	Cancer detection at the tile level	AUC: 0.979
					GS at the tile level	AUC: 0.899
					GP at the tile level	$\bar{F1}$: 0.65 (0.75 prev. Paper) qKappa: 0.655
Bulten, 2022 [76]	Evaluation of multiple algorithms (PANDA challenge)	10,616 WSIs	545 WSIs	741 patients (EV1) 330 patients (EV2)	GG classification	qKappa: 0.868 (EV2) qKappa: 0.862 (EV1)
Li, 2018 [68]	Multi-scale U-Net	187 tiles from 17 patients	37 tiles from 3 patients	None	Segment Stroma, benign and malignant gland segmentation	IoU: 0.755 (0.750 classic U-Net)
					Stroma, benign and GP 3/4 gland segmentation	IoU: 0.658 (0.644 classic U-Net)
Li, 2019 * [29]	R-CNN	513 WSIs	5 fold cross val	None	Stroma, benign, low- and high-grade gland segmentation	$\bar{IoU}$: 0.79 (mean amongst classes)
Bulten, 2019 * [28]	U-Net	62 WSIs	40 WSIs	20 WSIs	Benign vs. GP	IoU: 0.811 (IV) & 0.735 (EV) F1: 0.893 (IV) & 0.835 (EV)
Lokhande, 2020 [69]	FCN8 based on ResNet50	172 TMAs	72 TMAs	None	Benign, grade 3/4/5 segmentation	Dice: 0.74 (average amongst all classes)
Li, 2018 [71]	Multi-Scale U-Net-based CNN	50 patients	20 patients	None	Contribution of EM for multi-scale U-Net improvement	$\bar{IoU}$: 0.35 (U-Net) $\bar{IoU}$: 0.49 (EM-adaptative 30%)
Bulten, 2020 [41]	Extended Unet	5209 biopsies from 1033 patients	550 biopsies from 210 patients	886 cores	Malignancy detection at the WSI level	AUC: 0.99 (IV) & 0.98 (EV)
					GG > 2 detection	AUC: 0.978 (IV) & 0.871 (EV)
			100 biopsies	None	GG classification	qKappa: 0.819 (general pathologists) & 0.854 (DL on IV) & 0.71 (EV)
Hassan, 2022 [70]	ResNet50	18,264 WSIs	3251 WSIs	None	Tissue segmentation for GG presence	$\bar{IoU}$: 0.48 $\bar{F1}$: 0.375
Lucas, 2019 [67]	Inception V3	72 WSIs	24 WSIs	None	Malignancy detection at the pixel level	Sens: 0.90 Spec: 0.93
					GP3 & GP4 segmentation at pixel level	Sens: 0.77 Spec: 0.94

* Articles already in Table 4 for segmentation tissue performances. Double line separates classification (above) from segmentation algorithms (below). IV: Internal Validation. EV: External Validation. TMA: Tissue MicroArray. CNN: Convolutional Neural Network. GG: Gleason ISUP Group, GS: Gleason Score, GP: Gleason Pattern. AUC: Area Under the Curve. ACC: Accuracy. F1: F1-score, combination of precision and recall. IoU: Intersection over Union. q/wKappa: quadratic/weighted Cohen Kappa. Sens: Sensitivity. Spec: Specificity. $\bar{metric}$ implies the mean of this metric (e.g., $\bar{AUC}$).

### 3.3. Prediction

Deep learning was also used to predict clinical outcomes such as recurrence status, survival or metastasis (*n* = 10, see Table 7) or to predict genomic signatures from WSIs (*n* = 5, see Table 8). This is the most complex task to perform as no visible phenotypes are known by pathologists to make such decisions.

#### 3.3.1. Clinical Outcome Prediction

When focusing on recurrence, AUC around 0.8 [80,81] and Hazard Ratios (HR) above 4.8 [42,82,83] were obtained. A couple of articles studied the probability of developing metastasis [84,85]. The first article aimed to study if a patient developed lymph node metastasis after treatment [84] within undisclosed time frame, achieving an AUC of 0.69. The second article focused on distant metastasis, obtaining AUCs of 0.779 and 0.728 for 5- and 10-year metastasis. By combining image features and clinical data, the performance was improved to reach an AUC of 0.837 for 5- and 0.781 for ten years [85]. Liu et al. proposed to detect if benign slides belonged to a man who had no cancer or one who had cancer but on other biopsies and reached 0.74 AUC [86]. In any case, DL allows to establish decent survival models with HR of 1.38 for Nagpal et al. [73], 1.65 for Leo et al. [8] and 7.10 for Ren et al. [73,87,88].

**Table 7 diagnostics-13-02676-t007:** Articles focusing on clinical outcome prediction.

First Author, Year Reference	DL Architecture	Training Cohort	IV Cohort	EV Cohort	Aim	Results
Kumar, 2017 [81]	CNNs	160 TMAs	60 TMAs	None	Nucleus detection for tile selection	ACC: 0.89
					Recurrence prediction	AUC: 0.81(DL) & 0.59 (clinical data)
Ren, 2018 [83]	AlexNet + LSTM	271 patients	68 patients	None	Recurrence-free survival prediction	HR: 5.73
Ren, 2019 [88]	CNN + LSTM	268 patients	67 patients	None	Survival model	HR: 7.10 when using image features
Leo, 2021 [87]	Segmentation-based CNNs	70 patients	NA	679 patients	Cribriform pattern recognition	Pixel TPV: 0.94 Pixel TNV: 0.79
					Prognosis classification using cribriform area measurements	Univariable HR: 1.31 Multivariable HR: 1.66
Wessels, 2021 [84]	xse_ResNext34	118 patients	110 patients	None	LNM prediction based on initial RP slides	AUC: 0.69
Esteva, 2022 [85]	ResNet	4524 patients	1130 patients	None	Distant metastasis at five years (5Y) and ten years (10Y)	AUC: 0.837 (5Y) AUC: 0.781 (10Y)
					Prostate cancer-specific survival	AUC: 0.765
					Overall survival at ten years	AUC: 0.652
Pinckaers, 2022 [82]	ResNet50	503 patients	182 patients	204 patients	Univariate analysis for DL predicted biomarker evaluation	OR: 3.32 (IV) HR: 4.79 (EV)
Liu, 2022 [86]	10-CNN ensemble model	9192 benign biopsies from 1211 patients	2851 benign biopsies from 297 patients	None	Cancer detection at the patient level	AUC: 0.727
					Cancer detection at patient level from benign WSIs	AUC: 0.739
Huang, 2022 [80]	NA	243 patients	None	173 patients	Recurrence prediction at three years	AUC: 0.78
Sandeman, 2022 [42]	Custom CNN (AIForIA)	331 patients	391 patients	126 patients	Malignant vs. benign	AUC: 0.997
					Grade grouping	ACC: 0.67 wKappa: 0.77
					Outcome prediction	HR: 5.91

IV: Internal Validation. EV: External Validation. TMA: Tissue MicroArray. CNN: Convolutional Neural Network. RP: Radical Prostatectomies. LSTM: Long-Short Term Memory network. TMA: Tissue MicroArray. AUC: Area Under the Curve. ACC: Accuracy. wKappa: weighted Cohen Kappa. HR: Hazard Ratio. OR: Odds Ratio. TPV: True Positive Value. TNV: True Negative Value. $\bar{metric}$ implies the mean of this metric (e.g., $\bar{AUC}$).

#### 3.3.2. Genomic Signatures Prediction

Three groups started to work on the inference of genomic signatures from WSIs with the assumption that morphological features can predict pathway signatures [89,90,91]. These exploratory studies found correlations between RNA prediction from H&E images and RNA-seq signature expressions (PC ranging from 0.12 to 0.74). Schaumberg et al. carefully selected tiles containing tumor tissue and abnormal cells to train a classifier to predict SPOP mutations, reaching an AUC of 0.86 [92]. Dadhania et al. also used a tile-based approach to predict ERG-positive or negative mutational status, reaching around 0.8 AUC [93].

**Table 8 diagnostics-13-02676-t008:** Articles focusing on genomic signatures prediction.

First Author, Year Reference	DL Architecture	Training Cohort	IV Cohort	EV Cohort	Aim	Results
Schaumberg, 2018 [92]	ResNet50	177 patients	None	152 patients	SPOP mutation prediction	AUC: 0.74 (IV) AUC: 0.86 (EV)
Schmauch, 2020 [89]	HE2RNA	8725 patients (Pan-cancer)	5-fold cross val	None	Prediction of gene signatures specific to prostate cancer	PC: 0.18 (TP63) 0.12 (KRT8 & KRT18)
Chelebian, 2021 [91]	CNN from [37] fine-tuned	Pre-trained ([37])	7 WSIs	None	Correlation between clusters identified with AI and spatial transcriptomics	No global metric
Dadhania, 2022 [93]	MobileNetV2	261 patients	131 patients	None	ERG gene rearrangement status prediction	AUC: 0.82 to 0.85 (depending on resolution)
Weitz, 2022 [90]	NA	278 patients	92 patients	None	CCP gene expression prediction	PC: 0.527
					BRICD5 expression prediction	PC: 0.749
					SPOPL expression prediction	PC: 0.526

IV: Internal Validation. EV: External Validation. CNN: Convolutional Neural Network. AI: Artificial Intelligence. CCP: Cell Cycle Progression. AUC: Area Under the Curve. PC: Pearson Correlation.

### 3.4. Risk of Bias Analysis

A Risk of Bias (RoB) analysis was performed for every article. Details are described in Appendix A. Results are summarized in Figure 3. Missing criteria were categorized as high risk, and partially addressed criteria (e.g., only half the dataset is publicly available) were considered as intermediate risk. Articles that validated existing algorithms or focused on prediction algorithms where ground truth was not defined by people were classified as Not Applicable (NA). This analysis particularly outlines the lack of publicly available code (and hyperparameters) and data. External cohorts for validation are often not addressed. More efforts could be provided for model explainability and dealing with imbalanced data. Indeed, this is a common difficulty in pathomics that biases training and evaluation.

## 4. Discussion

### 4.1. Summary of Results

This review article provided a systematic search following PRISMA guidelines using multiple keywords in PubMed and Embase to identify studies related to DL for PCa in pathomics. The databases and the chosen keyword might represent limitations to this systematic review. Along with breast and colorectum, prostate cancer is the organ most explored by AI. It results in a high number of publications increasing every year (see Figure 4). Among them, nine articles focused on pre-processing steps, which are key to having a robust model, namely using normalization, quality assessment and smart tile selection. It is a rather new subject (see Figure 4), and its investigation is still ongoing, with new methods to improve how input data is handled. Among the 77 articles in this review, most algorithms (*n* = 50) were developed for the diagnosis of cancer and the Gleason grading. There are many different algorithms, and they tend to suffer the same biases (see Appendix A). It is, therefore, harder to see what the benefits from the models are compared to the others in their respective tasks. For detection purposes, classification algorithms were more common than segmentation algorithms. The former provided heat maps giving information on the location, while the latter provided precise information at the pixel level. For the diagnosis, the AUC comprised 0.77 and 0.99 with many papers (*n* = 16) with AUC above 0.95, favoring the use in routine activity.

The most performing algorithms used the MIL paradigm, allowing some explainability, and were trained on a large number of images. Algorithms for Gleason grading were less performant, with a quadratic Cohen Kappa comprised between 0.60 and 0.90. However, the definition of ground truth in Gleason grading suffers from high inter-observer variability that renders training less reliable. The prediction was less explored, with few articles approaching the prognosis (recurrence or metastasis development, for instance) or genomic alterations, but the interest in these investigations is increasing (see Figure 4). The AUC was comprised between 0.59 and 0.86. Prediction studies aim at correlating morphological patterns to a prediction that could enable the discovery of new patterns and help in patient personalized treatment. However, more robust studies, properly designed and validated, are needed to validate this assumption.

### 4.2. Biases

To evaluate the biases in our review, we adapted the QUADAS-2 tool to all the studies mentioned. Several biases can be noted in the studies at the image or algorithm level. Indeed, there is no standardization on the staining protocols, which translates to WSIs. This bias could be overcome using normalization such as Vahadane [94] or Macenko methods [95] or GANs (Generative Adversarial Networks) [24] or color augmentation [32,36,61,65,66,73,96], but most articles do not use image normalization to overcome this bias (50 out of 77 do not see Figure 3). In addition, most scanners have their proprietary format, which can impair the generalizability of the algorithms. Also, the brand of the scanner impacts the scanning process, potentially decreasing algorithms’ performance when trained on different brands. It is important to develop a strategy for testing algorithms with different scanners as proposed by the UK National Pathology Imaging Co-operative (NPIC). When working on diagnosis algorithms, ground truth is biased by inter-observer variability, especially in Gleason grading studies [21,41,77]. It is important to have multiple experts provide ground-truth annotation to not follow just one pathologist’s judgement, but it is not always the case (only 32 out of 77 articles do it, Figure 3).

Furthermore, there are a few general biases in the performance evaluation of algorithms. An imbalanced dataset will induce a bias if the issue is not addressed and not all articles consider it (only 41 out of 77, Figure 3). It is possible to use only a balanced dataset or DL techniques to reduce the impact of imbalance. A model should be trained multiple times to ensure its reproducibility. The performance average of all these trainings should be considered as the performance metric of the model. However, very few articles include confidence intervals on their metrics, which are yet key to evaluating the model in the existing state of the art. Less than half the articles include external validation cohorts (24 out of 77, Figure 3), but they are necessary to ensure that the evaluated model(s) is (are) not performing well only on the training WSIs, and that is also where normalization or color augmentation during training becomes crucial.

### 4.3. Limits of AI

There are limitations inherent to pathomics. Contrary to radiologic images, WSI has to be divided into tiles. Most classification algorithms must have a fixed input size, generally a defined number of tiles. It means that a subset of the slide has to be selected, and the heterogeneous aspect of the tumor might not be considered. This is also affected by the choice of magnification under which the WSIs is tiled [31]. A few articles focus on new methodologies to handle this type of data [82,97]. There are also articles that suggest a smart tile selection to use more informative data and reduce computational time [25]. The particularity in PCa pathology is that the main type of images are biopsies that contain a low percentage of tissue. It can be interesting to include multiple biopsies of patients to increase the number of tiles available for training. However, the way these different biopsies are given to algorithms has to be considered. At the very least, they have to be split into the same datasets (training or testing). Otherwise, bias could be induced in the study.

Ethical regulation makes access to data difficult [98]. The existence of challenges (e.g. PANDA, Prostate cANcer graDe Assessment) is a very good way to provide data to many researchers. It also facilitates collaborations on model development. It is necessary to be able to reproduce results, which is limited by the lack of publicly available cohorts (21 out of 77 used shared data, Figure 3). However, few publications shared completely the methodology and their code with consequences on the reproducibility of the model, hindering a proper comparison of usefulness and improvement of new algorithms (only 15 out of 77 shared it, Figure 3). The popularity of AI in these last years has also increased the number of models and data to be compared, computed, and stored. This has an economic but also environmental cost that needs to be addressed [99]. Computational costs can be reduced by using more frugal but efficient architectures. Transfer learning can also reduce training time using previously developed and trained models that are fine-tuned to fit the studied data. The downfall is to conform to the input data format. Focusing on more efficient architectures and how to properly share methodology in the field are potential improvements to be found to develop long-term viable solutions. The focus should also be directed towards the explainability of the algorithms. If they are to be implemented in clinical setups, “blackbox” models will not be trusted by pathologists (Figure 3, 40 of 77 attempted some form of explainability).

### 4.4. Impact of AI in Routine Activity

Nonetheless, several AI tools are now available on the market for the diagnosis in routine activity for prostate cancer from different companies: Ibex, Paige, Deep Bio and AIFORIA, whose algorithms were recently published [36,42,49,75]. They can be approved for first or second-read applications if they are used before as a screening or after as a quality check for the pathologist diagnosis. The Galen^TM^ Prostate from Ibex was the first to obtain CE under IVDR (In Vitro Diagnostic Medical Devices Regulation) in February 2023. The sensitivity and specificity of these products are very high when excluding slides with intermediate classification probability, also called undetermined categories. Indeed, a number of slides with undetermined categories are impacted by many parameters, such as pre-analytic conditions and the format of slides….

Consequently, performances depend on the site where the algorithms are deployed. In addition, their integration into routine activity supposes a digital workflow that is not widely available. Properly integrated into the workflow, it could help save time, but it is difficult to implement due to interoperability issues (integration in the Laboratory Information System (LIS) and the Image Management System (IMS)). An optimized integration supposes at least the automatized assignment of cases, the contextual launch, the cases prioritization, the visualization of heatmaps directly in the IMS and the integration of results directly in the report. Ethical considerations become an additional question when processing patient data, especially if sent to a cloud environment.

### 4.5. Multimodal Approach for Predictive Algorithms

Like other organs, the prediction of prostate cancer seems to be a more difficult question than detection. Indeed, the underlying assumption is that there exists a morphologic pattern in the images that can predict prognosis or genomic alteration. It is very likely that the answer is multifactorial and could benefit from multimodal approaches such as combining the WSI with radiologic, biological, and molecular data. The main challenge is properly combining all these data of different natures and evaluating the added value when combining them compared to the performance obtained by considering each separately.

## 5. Conclusions

In conclusion, DL has been widely explored in PCa, resulting in many pre-processing, diagnosis, or prediction publications. This systematic review highlights how DL could be used in this field and what significant improvements it could bring. It also included suggestions to reduce research biases in this field while outlining the inherent limits of these tools. Despite these limitations, PCa was one of the first organs to benefit from reliable AI tools that could already be used in routine activity for diagnosis purposes: cancer detection and Gleason grading… However, for predictive purposes, further studies are needed to improve the robustness of the algorithms, which could lead to more personalized treatment: prognosis, molecular alteration, etc.

## Figures and Tables

**Figure 1 diagnostics-13-02676-f001:**
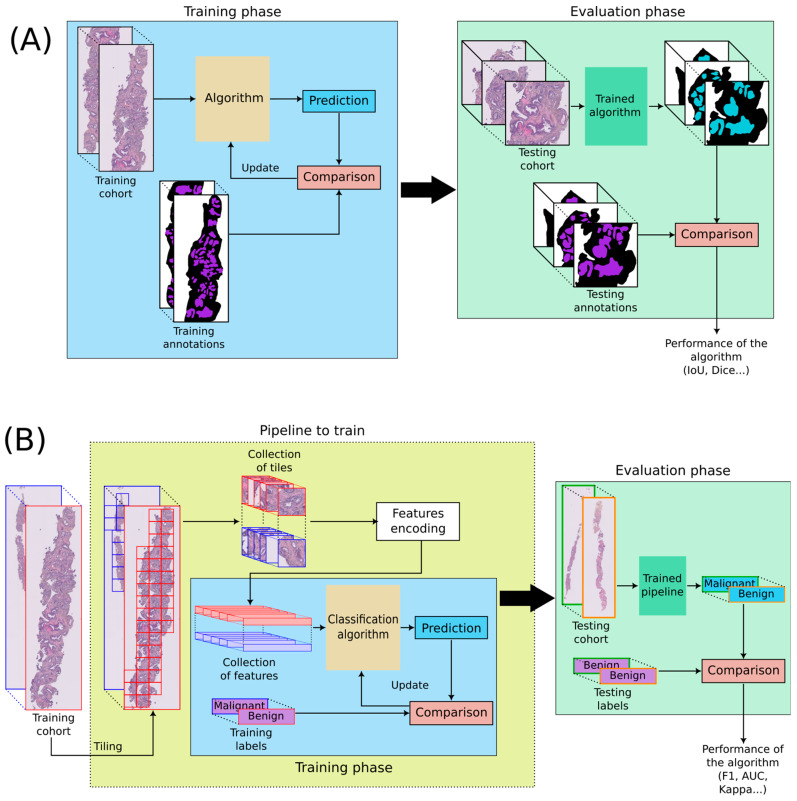
DL applied to WSIs. (**A**) Segmentation algorithm, (**B**) Classification algorithm. WSIs are divided into many tiles, and every tile is encoded into features. Tiling can be performed at different magnifications, but an identical number of tiles per WSIs is generally required. The encoding into features can be trained or performed with a pre-trained algorithm. Features are then used to train a classification algorithm.

**Figure 2 diagnostics-13-02676-f002:**
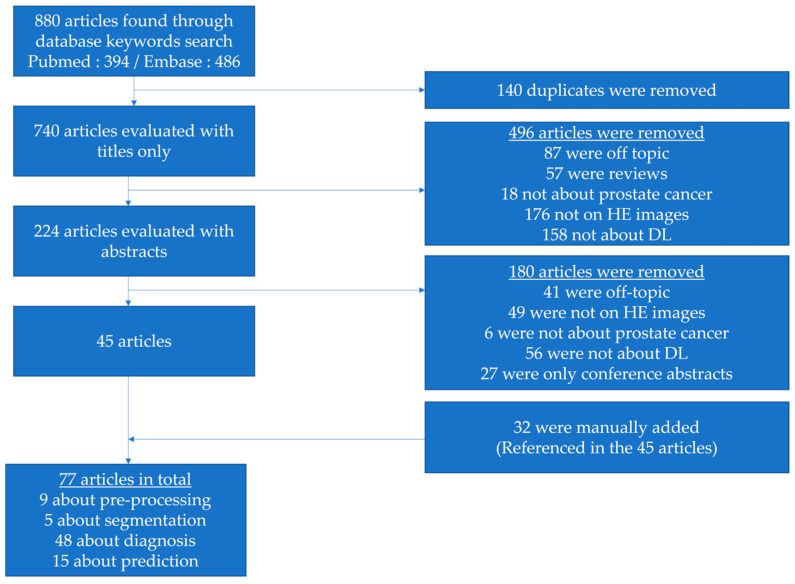
Flowchart illustrating the selection process of articles in this review.

**Figure 3 diagnostics-13-02676-f003:**
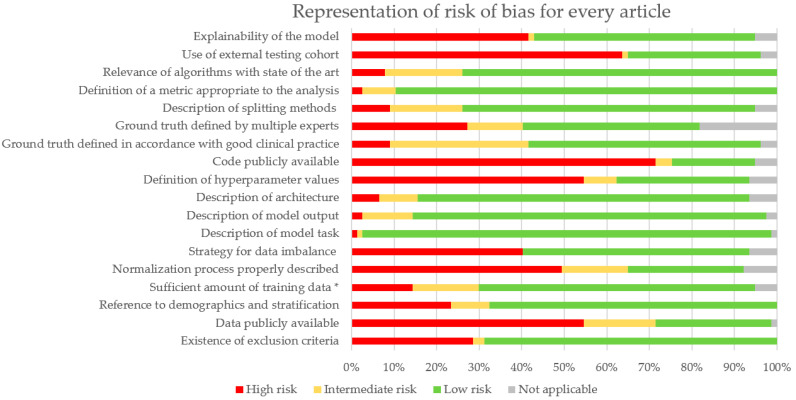
Proportion of RoB for all articles. * Sufficient amount was estimated at 200 WSIs for prediction and diagnosis tasks.

**Figure 4 diagnostics-13-02676-f004:**
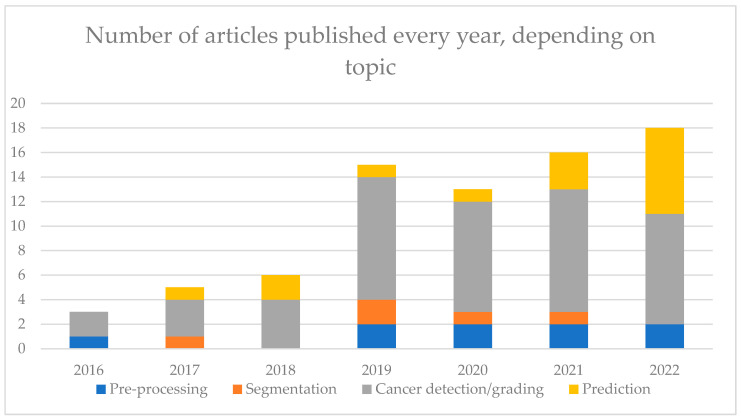
Number of articles on each topic, separated by year of publication.

**Table 1 diagnostics-13-02676-t001:** Correspondence between Gleason score and ISUP Gleason group.

Gleason ISUP Group	Gleason Score
1	Score = 6
2	Score 7 (3 + 4)
3	Score 7 (4 + 3)
4	Score 8 (4 + 4, 5 + 3, 3 + 5)
5	Score 9 or 10

**Table 2 diagnostics-13-02676-t002:** Checklist used to perform RoB analysis for every paper.

Quadas-2 Category	Adaptation	Criteria
Patient selection	Data source	Existence of exclusion criteria
		Data publicly available
		Reference to demographics and stratification
		Sufficient amount of training data *
	Data process	Normalization process properly described
		Strategy for data imbalance
Index Test	Model	Description of model task
		Description of model output
		Description of architecture
		Definition of hyperparameter values
		Code publicly available
Reference standard	Ground truth	Ground truth defined in accordance with good clinical practice
		Ground truth defined by multiple experts
Flow and timing	Analysis	Description of splitting methods
		Definition of a metric appropriate to the analysis
		Relevance of algorithms with state of the art
		Use of external testing cohort
		Explainability of the model

* For diagnosis and prediction tasks, a number of 200 WSIs was required.

**Table 3 diagnostics-13-02676-t003:** Papers focusing on pre-processing tasks.

First Author, Year Reference	DL Architecture	Training Cohort	IV Cohort	EV Cohort	Aim	Results
Schömig-Markiefka, 2021 [16]	InceptionResNetV2	Already trained	Subset of TCGA slides	686 WSIs	Impact of artifacts on tissue classification	Tissue classification performance decreases with the appearance of artifacts
Haghighat, 2022 [17]	ResNet18 PathProfiler	198 WSIs	3819 WSIs	None	Prediction of tile-level usability	AUC: 0.94 (IV)
					Prediction of slide-level usability	AUC: 0.987 (IV) PC: 0.889 (IV)
Brendel, 2022 [18]	ResNet34-IBN	4984 WSIs TCGA (6 cancer types)	866 WSIs TCGA (6 cancer types)	78 WSIs	Cancer and tissue type prediction	F1: 1 (IV) F1: 0.83 (EV)
					Tumor purity above 70%	AUC: 0.8 (IV)
Anghel, 2019 [19]	VGG-like network	96 WSIs	14 WSIs	None	Improvement of cancer detection when using poor-quality WSIs	F1 (base): 0.79 F1 (best): 0.87
Otàlora, 2019 [20]	MobileNet [21] + 2 layers	3540 TMAs	1321 TMAs	3961 TMAs	Impact of normalization on GP classification	AUC (base): 0.56 (IV) & 0.48 (EV) AUC (best): 0.84 (IV) & 0.69 (EV)
Rana, 2020 [22]	GANs	148 WSIs	13 WSIs	None	Comparison of dye and computationally stained images	$\bar{PC}$: 0.96 Structural similarity index: 0.9
					Comparison of unstained and computationally destained images	$\bar{PC}$: 0.96 Structural similarity index: 0.9
Sethi, 2016 [23]	Custom CNN	20 WSIs	10 WSIs	None	Epithelial-stromal segmentation	AUC: 0.96 (Vahadane) AUC: 0.95 (Khan)
Swiderska-Chadaj, 2020 [24]	U-Net GANs	324 WSIs	258 WSIs	85 WSIs	Impact of normalization on cancer detection	AUC: 0.92 AUC: 0.98 (GAN)
				50 WSIs		AUC: 0.83 AUC: 0.97 (GAN)
Salvi, 2021 [25]	Inceptionv3 pre-trained on ImageNet	400 WSIs	100 WSIs	None	WSI-level classification algorithm performance adding normalization and tile selection	Sens (base): 0.94 Spec (base): 0.68 Sens (best): 1 Spec (best): 0.98

IV: Internal Validation. EV: External Validation. TCGA: The Cancer Genome Atlas project. TMA: Tissue Micro Array. CNN: Convolutional Neural Network. GP: Gleason Pattern. AUC: Area Under the Curve. PC: Pearson Correlation. F1: F1-score. $\bar{metric}$ implies the mean of this metric (e.g., $\bar{PC}$).

**Table 4 diagnostics-13-02676-t004:** Papers focusing on segmenting glands or tissue to help future classification tasks.

First Author, Year Reference	DL Architecture	Training Cohort	IV Cohort	EV Cohort	Aim	Results
Ren, 2017 [26]	U-Net	22 WSIs	5-fold cross val	None	Gland Segmentation in H&E slides	F1: 0.84
Li, 2019 [29]	R-CNN	40 patients	5-fold cross val	None	Epithelial cell detection	AUC: 0.998
Bulten, 2019 [28]	U-Net	20 WSIs	5 WSIs	None	Epithelium segmentation based on IHC	IoU: 0.854 F1: 0.915
Bukowy, 2020 [30]	SegNet (VGG16)	10 WSIs	6 WSIs	None	Weakly- and strongly-annotated segmentation	$\bar{AUC}$: 0.85 (strong) $\bar{AUC}$: 0.93 (weak fine-tuned strong)
		140 WSIs			Epithelium segmentation with a combination of 3 models	$\bar{ACC}$: 0.86 (DL)
Salvi, 2021 [27]	U-Net + post-processing	100 patients	50 patients	None	Gland segmentation	Dice: 0.901 U-Net + post-processing Dice: 0.892 U-Net only

IV: Internal Validation. EV: External Validation. CNN: Convolutional Neural Network. AUC: Area Under the Curve. F1: F1-score. IoU: Intersection Over Union. $\bar{metric}$ implies the mean of this metric. cross val: cross-validation.

**Table 5 diagnostics-13-02676-t005:** Papers focusing on cancer detection only.

First Author, Year Reference	DL Architecture	Training Cohort	IV Cohort	EV Cohort	Aim	Results
Litjens, 2016 [43]	Custom CNN	150 patients	75 patients	None	Cancer detection at the pixel level	AUC: 0.99
Kwak, 2017 [44]	Custom CNN	162 TMAs	491 TMAs	None	Cancer detection at the sample level	AUC: 0.974
Kwak, 2017 [45]	Custom CNN	162 TMAs	185 TMAs	None	Cancer detection at the sample level	AUC: 0.95
Campanella, 2018 [46]	ResNet34 and VGG11-BN (MIL)	12610 WSIs	1824 WSIs	None	Cancer detection at the WSI level	AUC: 0.98
Campanella, 2019 [32]	RNN (MIL)	12 132 WSIs	1784 WSIs	12 727 WSIs	Cancer detection at the WSI level	AUC: 0.991 (IV) & 0.932 (EV)
	ResNet34 (MIL)					AUC: 0.986 (IV)
Garcià, 2019 [55]	VGG19	6195 glands (from 35 WSIs)	5-fold cross val	None	Malignancy gland classification	AUC: 0.889
Singh, 2019 [50]	ResNet22	749 WSIs	3-fold cross val	None	Cribriform pattern detection at the tile level	ACC: 0.88
Jones, 2019 [56]	ResNet50 & SqueezeNet	1000 tiles from 10 WSIs	200 tiles from 10 WSIs	70 tiles from unknown number of WSIs	Malignancy detection at the tile level	$\bar{ACC}$: 0.96 (IV) & 0.78 (EV)
Duong, 2019 [31]	ResNet50 and multiscale embedding	602 TMAs	303 TMAs	None	TMA classification using only ×10 magnification	AUC: 0.961
					TMA classification with multi-scale embedding	AUC: 0.971
Raciti, 2020 [34]	PaigeProstate	Pre-trained [32]	Pre-trained [32]	232 biopsies	Malignancy classification at the WSI level	Sens: 0.96 Spec: 0.98
					Improvement of pathologist’s classification	$\bar{Sens}$: 0.738 to 0.900 $\bar{Spec}$: 0.966 to 0.952
Han, 2020 [38]	AlexNet	286 WSIs from 68 patients	Leave one out cross val	None	Cancer classification at the WSI level	AUC: 0.98
Ambrosini, 2020 [51]	Custom CNN	128 WSIs	8-fold cross val	None	Cribriform pattern detection for biopsies	$\bar{AUC}$: 0.8
Bukhari, 2021 [57]	ResNet50	640 WSIs	162 WSIs	None	Cancer/hyperplasia detection at the tile level	F1: 1 ACC: 0.995
Pinckaers, 2021 [33]	ResNet-34 (MIL)	5209 biopsies	535 biopsies	205 biopsies	Cancer detection for biopsies	AUC: 0.99 (IV) & 0.799 (EV)
	Streaming CNN					AUC: 0.992 (IV) & 0.902 (EV)
Perincheri, 2021 [49]	Paige Prostate	Pre-trained [34]	Pre-trained [34]	1876 biopsies from 116 patients	Paige classification evaluation	110/118 patients were correctly classified
Da Silva, 2021 [48]	PaigeProstate	Pre-trained [34]	Pre-trained [34]	600 biopsies from 100 patients	Malignancy classification for biopsies	Sens: 0.99 Spec: 0.93
					Malignancy classification for patients	Sens: 1 Spec: 0.78
Raciti, 2022 [47]	PaigeProstate	Pre-trained [34]	Pre-trained [34]	610 biopsies	Cancer detection for patients	Sens: 0.974 Spec: 0.948 AUC: 0.99
Krajnansky, 2022 [58]	VGG16-mode	156 biopsies from 262 WSIs	Ten biopsies from 87 WSIs	None	Malignancy detection for biopsies	FROC: 0.944
					Malignancy detection for patients	AUC: 1
Tsuneki, 2022 [53]	EfficientNetB1 pre-trained on colon	1182 needle biopsies	1244 TURP biopsies 500 needle biopsies	767 WSIs	Cancer detection for classic and TURP biopsies	AUC: 0.967 (IV) & 0.987 (EV) AUC (TURP): 0.845
	EfficientNetB1 pre-trained on ImageNet					AUC: 0.971 (IV) & 0.945 (EV) AUC (TURP): 0.803
Tsuneki, 2022 [54]	EfficientNetB1 pre-trained on ImageNet	1060 TURP biopsies	500 needle biopsies, 500 TURP	768 WSIs	Cancer detection in classic and TURP biopsies	AUC: 0.885 (IV) TURP AUC: 0.779 (IV) & 0.639 (EV) biopsies
	EfficientNetB1 pre-trained on colon					AUC: 0.947 (IV) TURP AUC: 0.913 (IV) & 0.947 (EV) biopsies
Chen, 2022 [59]	DenseNet	29 WSIs	3 WSIs	None	Classification of tissue malignancy	AUC: 0.98 (proposed method) AUC: 0.90 (DenseNet-121)

IV: Internal Validation. EV: External Validation. TMA: Tissue MicroArray. CNN: Convolutional Neural Network. MIL: Multiple Instance Learning. TURP: TransUrethral Resection of Prostate. AUC: Area Under the Curve. FROC: Free Receiver Operating Characteristic. ACC: Accuracy. F1: F1-score. Sens: Sensitivity. Spec: Specificity. $\bar{metric}$ implies the mean of this metric (e.g., $\bar{AUC}$).

## Data Availability

The data presented in this study are available in the Appendix A.

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
