# Peer review of "Deep Learning Methodologies Applied to Digital Pathology in Prostate Cancer: A Systematic Review"

_diagnostics, 2023, doi:10.3390/diagnostics13162676_

Round 1

Reviewer 1 Report

This systematic review explores the application and performance of deep learning algorithms in digital pathology for prostate cancer. The review collected relevant articles using Pubmed and Embase and assessed the Risk of Bias using an adaptation of the QUADAS-2 tool. My comments are listed below: 

1) Additional papers could be included on pre-processing and patch selection (doi: 10.1117/12.2255710; 10.1016/j.cmpbup.2021.100004), the deep learning portion (doi: 10.1117/12.2512807), and the comparison of AI grading versus pathologists (doi: 10.1016/j.compbiomed.2023.106856);

2) The 'Risk of Bias Analysis' section is highly appreciated. I would expect the authors to discuss in more detail the results that emerged from the analysis (Figure 3) in the discussion section.

3)Summary of the results: It would be helpful to have a more in-depth discussion of the main findings of this review by the experts. Additionally, I suggest including the number of articles published each year for each topic (pre-processing, segmentation, classification/grading, prediction) to give readers an idea of the level of interest in each area.

4) Are there any significant aspects that the authors want to delve into in the discussion regarding this topic? What considerations can be made about AI tools applied to prostate pathology? What are the issues related to the use of these tools in routine clinical practice? 

Author Response

Dear reviewer,

Thank you again for your comments and suggestions, please see the attachement for our response.

Respectfully,

Noémie Rabilloud for all authors.

Reviewer 2 Report

1. Figure 1A and 1B are wrongly addressed as 2A & 2B.

2. Give details on the pipelined architecture ( given in figure 2B), for handling the input.

3. Provide significant contributions with clear details

4. Lines 102 to 106, need reorganization and details

5.  Discuss the utilization of  the transfer learning models and customized models  regarding

6. Conclusion section writeup needs to be improved, by elaborating the section. The focus may have to be given to  highlight the  systematic study, and the emphasis on the outcome and the future benefit

Author Response

(The authors gave the same response as above.)

Round 2

Reviewer 1 Report

The authors addressed all my previous comments